organic chemistry/synthetic chemistry/ spectroscopy

efficient, synthesis, furan-3(2H)-imine, scaffold, alkynones, authors

**Authors for correspondence:**
Anas J. Rasras
e-mail: rasras.chem@bau.edu.jo
Ihsan A. Shehadi
e-mail: ishehadi@sharjah.ac.ae

# An efficient synthesis of furan-3(2H)-imine scaffold from alkynones

Anas J. Rasras[1], Ihsan A. Shehadi[2], Eyad A. Younes[3], Da'san M. M. Jaradat[1] and Raed A. AlQawasmeh[2]

[1]Faculty of Science, Department of Chemistry, Al-Balqa Applied University, PO Box 19117, Al-Salt, Jordan
[2]College of Science, Department of Chemistry, University of Sharjah, Pure and Applied Chemistry Research Group, PO Box 27272, Sharjah, United Arab Emirates
[3]Department of Chemistry, Faculty of Science, The Hashemite University, PO Box 330127, Zarqa 13133, Jordan

IAS, 0000-0001-5327-4601

A novel efficient method to generate spiro furan-3(2H)-imine derivatives is established by the reaction between the $\alpha,\beta$-unsaturated ketones and aniline derivatives. The reaction involves 1,4- addition of aniline followed by the subsequent intramolecular cyclization mediated by tertiary alcohol to produce the furan-3(2H)-imine. All the synthesized compounds are characterized using nuclear magnetic resonance and high-resolution mass spectrometry (HRMS).

## 1. Introduction

Furanone occupies an important area in heterocyclic chemistry because it has a large spectrum of biological activities such as anti-cancer [1,2], anti-inflammatory [3] and antibiotic [4]. In addition, furanone is highly appreciated by the food industry as it is considered a key flavour compound in many fruits [5].

Over many decades, imine moiety has been receiving increasing attention due to a wide range of biological activities as well like anti-cancer [6], antioxidant [7], antimalarial [8], antibacterial [9,10] and anti-HIV [11]. Recently, imine- nanocapsules show a highly effective delivery of an insufficiency soluble cancer drug into tumour cells [12]. Organic imine cages also show significant applications in the detection of toxic organic pollutants [13].

The synthesis of a hybrid scaffold that contains furan-3(2H)-imine from furan-3(2H)-one and amines is very challenging due to the poor electrophilicity of carbonyl group in furan-3(2H)-one derivatives. Recently, Chen *et al.* reported the synthesis of furan-3(2H)-imine through the reaction of β-enamino-α,β-unsaturated ketone with a base. However, this synthetic strategy is limited to β-difluoro-β-enamioketone (scheme 1) [14]. Shang *et al.*

**Scheme 1.** Reaction of amine with α,β-unsaturated ketone.

**Scheme 2.** Synthesis of 4-hydroxybut-2-ynone **4(a-d)**.

developed a method to synthesis 4-arylsulfonylimino-4,5-dihydrofurans, using CuI catalyst and three-component reaction by mixing sulfonyl azide, phenylacetylene and β-ketoester (scheme 1) [15]. Furanylidenamines derivatives are synthesized from dieneamides using HCl [16], silver [17], palladium [18–21], copper [22] and catalyst. Other methods are published on the preparation of furanylidenamines via the reaction between isocyanide and alkyne [23–25]. The development of new synthetic strategies that provide access to new biologically active compounds is highly required. Because of this, we present in this work a new and efficient method for the synthesis of new furan-3(2H)-imine derivatives.

## 2. Results and discussion

The treatment of propargylic alcohol derivatives **2(a-b)** with *n*-BuLi at −78°C generates the deprotonated anion intermediate, where the resulting anion *in situ* reacts with benzaldehyde derivatives **1(a-c)** to offer but-2-yne-1,4-diol derivatives **3(a-d)** in 73–85% yield (scheme 2) [26]. The precursor α, β-unsaturated ketone derivatives **4(a-d)** are achieved in 64–84% yield by the oxidation of diols **3(a-d)** under mild condition using $MnO_2$ (scheme 2) [27]. By having compounds **4(a-d)** in hand, the synthesis of the target compounds **6(a-t)** has been achieved in a one-step reaction through heating of aniline derivatives and 4-hydroxybut-2-ynone **4(a-d)** in MeCN:$H_2O$ (1:1) for 0.5–3 h. The desired products **6(a-t)** are obtained in good to excellent yield as are shown in table 1. The geometry of the C=N double bonds in the products has been concluded based on the associated steric effects.

**Table 1.** Synthesis of furan-3(2*H*)-imine **6**(**a-t**).

It is clear from the obtained results that the reaction of 4-aminophenol with cyclohexyl moiety gives the best percentage yields, indeed, **6e**, **6j** and **6o** with 91%, 94% and 91%, respectively, and are produced in just 0.5 h for **6e** and **6j** (table 1).

Scheme 3 shows another approach where furan-3(2*H*)-one **5** is chosen as the building block for the synthesis of furan-3(2*H*)-imine through the condensation reaction with aniline derivatives. However,

**Scheme 3.** Attempt to synthesize furan-3(2H)-imine **6a** form furan-3(2H)-one **5**.

**Scheme 4.** Proposed mechanism of the cyclization reaction to form furan-3(2H)-imine.

there is not any significant amount of the target product being observed in such reaction, hence, the suggested reaction pathway is not favoured. The lack of reactivity of compound **5** can be associated with the electron-donating effects of oxygen atom in the β-position which can significantly reduce the electrophilicity of carbonyl group (scheme 3).

Scheme 4 shows a plausible mechanism for the final product formation, exemplified by compound **6a**. Herein, an initial 1,4-addition produces the intermediate zwitterions **A** that is followed by a hydrogen transfer to produce the enamine **B**. Prevalent isomerization of **B** produces the imine **C**, in which intramolecular nucleophilic attack of the tertiary alcohol to the carbonyl moiety produces the cyclic hydroxylfurane skeleton **D**. The intermediate **D** releases water to produce furan-3(2H) imine derivative **6a**.

Aniline derivatives with hydroxy group show excellent yields, while aniline with electron-withdrawing group such as 4-chloroaniline and 4-nitroaniline is observed in trace amounts. The purity of all products is identified by thin-layer chromatography (TLC) using different mobile phases with different polarities. All the synthesized analogues are characterized by $^1$H, $^{13}$C nuclear magnetic resonance (NMR) and high-resolution mass spectrometry (HRMS). The $^1$H NMR spectra of each **6(a-t)** show a significant peak at $\delta = 5.91$–6.50 ppm that can be associated with the CH at α-imine moiety. Each $^{13}$C NMR spectrum shows a signal at $\delta = 89.1$–92.6 ppm which is related to the spiro quaternary carbon and a peak at $\delta = 92.4$–93.9 ppm for the CH at α-imine moiety, while the peak at $\delta = 170.8$–174.3 ppm belongs to imine carbon. Compound **6o** is further characterized by DEPT 135, COSY, HMQC and HMBC as an example, the spectral data are fitted to the suggested structure. Figure 1 shows the significant correlations which are derived from these techniques for compound **6o**.

When the same reaction is performed with the 4-aminopyridine or hydroxyl amine, the only obtained product is furan-3(2H)-one 5 (scheme 5), which reflects the low stability of the imine product in both cases. The stability compound **6a** is examined in a basic medium by refluxing in 10% NaOH for 5 h, and in acidic medium by heating at 80°C in 5% H$_2$SO$_4$ for 24 h. These reactions show no changes on **6a** under these conditions indicating the high stability of these analogues.

**Figure 1.** HMBC correlations of compound **6o**.

**Scheme 5.** Trials to synthesize furan-3(2*H*)-imine.

# 3. Conclusion

Furan-3(2*H*)-imine analogues have been synthesized in one step by 1,4- addition of aniline derivatives to $\alpha,\beta$-unsaturated carbonyl moieties, which undergo intramolecular cyclization via tertiary spiro alcohol to produce the desired products. The structures of all newly synthesized compounds have been confirmed by NMR spectra and HRMS of representative analogues.

# 4. Experimental section

## 4.1. General

NMR spectra are measured with tetramethylsilane as the internal standard. NMR spectra are recorded on a Bruker Avance III-500 MHz spectrometer type ($^1$H NMR, 500 MHz; $^{13}$C NMR, 125 MHz). Chemical shifts values ($\delta$) are reported in ppm. HRMS are recorded in positive ion mode by electrospray ionization using a Bruker Daltonics Apex IV, 7.0T Ultra Shield Plus (Bremen, Germany). Melting points (mp) are determined on an electrothermal melting point apparatus and are uncorrected. Unless otherwise noted, all chemicals and solvents used are obtained from Fluka and Aldrich, and used without further purification. Tetrahydrofuran (THF) is distilled over sodium metal and benzophenone after reflux. All reactions are monitored by TLC using Merck aluminium plates pre-coated with silica gel PF254 ($20 \times 20 \times 0.2$ mm), and visual detection of the plate is observed under UV lamp ($\lambda = 254$ nm).

## 4.2. General procedure for the preparation of compound **3**(**a-d**)

Compounds **3**(**a-d**) are synthesized according to the literature procedure [28]. To a suspension of 1-ethynylalcohol **2** (24.16 mmol, 1.0 eq) in THF (100 ml) is added dropwise a 1.6 M solution of n-BuLi in n-hexane (48.32 mmol, 2.1 eq) at –78°C. After stirring for 30 min, aldehyde **1** (24.16 mmol, 1.0 eq) is added dropwise and the solution is allowed to warm to room temperature (r.t.). After 2 h, H$_2$O (100 ml) is added and the solution is extracted with CH$_2$Cl$_2$ ($3 \times 100$ ml). The combined organic layers are dried over MgSO$_4$, and the solvent is evaporated. The obtained diol is purified by column chromatography (SiO$_2$, hexanes–EtOAc, 4 : 1) to give diol **3**.

## 4.3. 1-(3-hydroxy-3-(p-tolyl)prop-1-yn-1-yl)cyclo hexan-1-ol (**3a**)

Colorless oil (5.02 g, 85% yield). $^1$H NMR (500 MHz, CDCl$_3$) $\delta$ 7.43 (d, $J$ = 7.9 Hz, 2H), 7.19 (d, $J$ = 7.8 Hz, 2H), 5.48 (d, $J$ = 5.4 Hz, 1H), 2.36 (s, 3H), 1.75–1.50 (m, 10H). $^{13}$C NMR (125 MHz, CDCl$_3$) $\delta$ 138.2, 137.8, 129.2, 126.6, 90.2, 84.0, 68.7, 64.4, 39.8, 25.1, 23.2, 21.1.

## 4.4. 1-(3-hydroxy-3-phenylprop-1-yn-1-yl)cyclo hexan-1-ol (**3b**)

White solid (4.51 g, 81% yield). mp 80–82°C. $^1$H NMR (500 MHz, CDCl$_3$) $\delta$ 7.64–7.47 (m, 2H), 7.42–7.29 (m, 3H), 5.47 (d, $J$ = 5.2 Hz, 1H), 3.54 (d, $J$ = 5.7 Hz, 1H), 3.14 (s, 1H), 2.03–1.83 (m, 2H), 1.78–1.43 (m, 8H). $^{13}$C NMR (125 MHz, CDCl$_3$) $\delta$ 140.6, 128.5, 128.2, 126.7, 90.3, 84.0, 68.7, 64.3, 39.7, 25.0, 23.2. HRMS (ESI-MS): $m/z$ calcd for C$_{15}$H$_{17}$O$_2$: 229.1234 [M-H]$^+$ is found to be 229.1211.

## 4.5. 1-(3-(4-chlorophenyl)-3-hydroxyprop-1-yn-1-yl)cyclohexan-1-ol (**3c**)

White solid (4.99 g, 78% yield). mp 122–124°C. $^1$H NMR (500 MHz, CDCl$_3$) $\delta$ 7.45 (d, $J$ = 8.3 Hz, 2H), 7.33 (d, $J$ = 8.4 Hz, 2H), 5.46 (s, 1H), 2.99 (s, 1H), 2.57 (s, 1H), 2.02–1.80 (m, 2H), 1.76–1.46 (m, 8H). $^{13}$C NMR (125 MHz, CDCl$_3$) $\delta$ 139.1, 134.1, 128.7, 128.0, 90.7, 83.5, 68.7, 63.7, 39.7, 25.0, 23.2.

## 4.6. 1,1-diphenyl-4-(p-tolyl)but-2-yne-1,4-diol (**3d**)

White solid (5.79 g, 73% yield). mp 178–180°C. $^1$H NMR (500 MHz, DMSO-d$_6$) $\delta$ 7.59–7.48 (m, 4H), 7.45–7.37 (m, 2H), 7.34–7.26 (m, 4H), 7.22–7.14 (m, 4H), 6.74 (s, 1H), 6.02 (d, $J$ = 6.0 Hz, 1H), 5.48 (d, $J$ = 6.0 Hz, 1H), 2.30 (s, 3H). $^{13}$C NMR (125 MHz, DMSO-d$_6$) $\delta$ 146.4, 139.2, 136.6, 128.6, 127.8, 126.9, 126.4, 125.6, 88.0, 87.3, 72.7, 62.4, 20.6. HRMS (ESI-MS): $m/z$ calcd for C$_{23}$H$_{20}$NaO$_2$: 351.1356 [M + Na]$^+$ is found to be 351.1352.

## 4.7. General procedure for the preparation of compound **4**

To a suspension of diol **3** (20.96 mmol, 1 eq) in CH$_2$Cl$_2$ (150 ml), an excess of MnO$_2$ (209.60 mmol, 10 eq) is added, the mixture is then stirred for 20 h at r.t. The suspension is then filtered through a pad of Celite, evaporated to dryness and the product is purified by column chromatography (SiO$_2$, hexanes–EtOAc, 6 : 1) to give ketone **4**. Note: compound **4d** is synthesized from compound **3d** and used in the next step without further purification.

## 4.8. 3-(1-hydroxycyclohexyl)-1-(p-tolyl)prop-2-yn-1-one (**4a**)

Yellow oil (4.27 g, 84% yield). $^1$H NMR (500 MHz, CDCl$_3$) $\delta$ 7.99 (d, $J$ = 8.1 Hz, 2H), 7.25 (d, $J$ = 8.1 Hz, 2H), 2.40 (s, 3H), 2.05–1.94 (m, 2H), 1.77–1.48 (m, 8H). $^{13}$C NMR (125 MHz, CDCl$_3$) $\delta$ 177.6, 145.3, 134.3, 129.7, 129.3, 96.9, 82.0, 68.8, 39.3, 24.9, 23.0, 21.8. HRMS (ESI-MS): $m/z$ calcd for C$_{16}$H$_{18}$NaO$_2$: 265.1199 [M + Na]$^+$ is found to be 265.1174.

## 4.9. 3-(1-hydroxycyclohexyl)-1-phenylprop-2-yn-1-one (**4b**)

Orange oil (3.40 g, 71% yield). $^1$H NMR (500 MHz, CDCl$_3$) $\delta$ 8.11 (d, $J$ = 7.4 Hz, 2H), 7.67–7.55 (m, 1H), 7.52–7.37 (m, 2H), 3.10 (s, 1H), 2.08–2.05 (m, 2H), 1.78–1.57 (m, 8H). $^{13}$C NMR (125 MHz, CDCl$_3$) $\delta$ 178.0, 136.5, 134.2, 129.5, 128.6, 97.8, 81.8, 68.7, 39.2, 24.9, 23.0.

## 4.10. 1-(4-chlorophenyl)-3-(1-hydroxycyclohexyl)prop-2-yn-1-one (**4c**)

Yellow solid (3.96 g, 72% yield). mp 63–65°C. $^1$H NMR (500 MHz, CDCl$_3$) $\delta$ 8.05 (d, $J$ = 8.5 Hz, 2H), 7.45 (d, $J$ = 8.5 Hz, 2H), 2.41 (s, 1H), 2.15–2.04 (m, 2H), 1.80–1.59 (m, 8H), 1.37–1.32 (m, 1H). $^{13}$C NMR (125 MHz, CDCl$_3$) $\delta$ 176.5, 140.8, 135.0, 130.8, 129.0, 98.0, 81.6, 68.8, 39.2, 24.9, 23.0.

## 4.11. General procedure for the preparation of compound **6**

To a ketone **4** (0.82 mmol, 1 eq) in a mixture of MeCN:H$_2$O (1 : 1, 15 ml), aniline **5** (0.90 mmol, 1.1 eq$_,$) is added, the mixture is then heated to 100°C. After cooling the mixture to r.t., H$_2$O (10 ml) is added and the solution is

extracted with $CH_2Cl_2$ ($3 \times 10$ ml). The combined organic layers are dried over $MgSO_4$, and the solvent is evaporated. Imine obtained is purified by TLC ($SiO_2$, hexanes–EtOAc, 4 : 1) to give imine **6**.

### 4.12. N-phenyl-2-(p-tolyl)-1-oxaspiro[4.5]dec-2-en-4-imine (6a)

Yellow oil (114.5 mg, 44% yield). $^1$H NMR (500 MHz, CDCl$_3$) $\delta$ 7.73–7.52 (m, 2H), 7.41–7.19 (m, 4H), 7.06–6.82 (m, 3H), 5.94 (s, 1H), 2.37 (s, 3H), 1.87–1.81 (m, 9H), 1.54–1.32 (m, 1H). $^{13}$C NMR (125 MHz, CDCl$_3$) $\delta$ 178.3, 173.7, 152.6, 141.6, 129.3, 128.8, 127.2, 126.6, 123.1, 121.3, 92.6, 90.1, 34.4, 24.7, 22.2, 21.6. HRMS (ESI-MS): $m/z$ calcd for $C_{22}H_{24}NO$: 318.1852 [M + H]$^+$ is found to be 318.1847.

### 4.13. N,2-di-p-tolyl-1-oxaspiro[4.5]dec-2-en-4-imine (6b)

Yellow solid (111.4 mg, 41% yield). mp 91–93°C. $^1$H NMR (500 MHz, DMSO-d$_6$) $\delta$ 7.73 (d, J = 8.1 Hz, 2H), 7.28 (d, J = 8.0 Hz, 2H), 7.12 (d, J = 7.9 Hz, 2H), 6.77 (d, J = 8.0 Hz, 2H), 6.18 (s, 1H), 2.35 (s, 3H), 2.28 (s, 3H), 1.93–1.55 (m, 9H), 1.47–1.28 (m, 1H). $^{13}$C NMR (125 MHz, DMSO-d$_6$) $\delta$ 176.4, 172.7, 149.3, 141.4, 131.6, 129.3, 129.2, 126.5, 126.4, 120.8, 92.4, 89.1, 34.1, 24.2, 21.6, 21.0, 20.4. HRMS (ESI-MS): $m/z$ calcd for $C_{23}H_{26}NO$: 332.2009 [M + H]$^+$ is found to be 332.1987.

### 4.14. N-(4-methoxyphenyl)-2-(p-tolyl)-1-oxaspiro[4.5]dec-2-en-4-imine (6c)

Yellow oil (165.3 mg, 58% yield). $^1$H NMR (500 MHz, CDCl$_3$) 7.63 (d, J = 8.1 Hz, 2H), 7.21 (d, J = 8.0 Hz, 2H), 6.99–6.82 (m, 4H), 6.01 (s, 1H), 3.81 (s, 3H), 2.39 (s, 3H), 1.86–1.74 (m, 9H), 1.53–1.35 (m, 1H). $^{13}$C NMR (125 MHz, CDCl$_3$) $\delta$ 178.1, 173.3, 155.8, 141.5, 129.2, 127.3, 127.1, 126.5, 122.3, 114.1, 92.6, 90.1, 55.4, 34.5, 24.7, 22.2, 21.6. HRMS (ESI-MS): $m/z$ calcd for $C_{23}H_{26}NO_2$: 348.1958 [M + H]$^+$ is found to be 348.1973.

### 4.15. N-(4-chlorophenyl)-2-(p-tolyl)-1-oxaspiro[4.5]dec-2-en-4-imine (6d)

Yellow solid (124.1 mg, 43% yield). mp 111–113°C. $^1$H NMR (500 MHz, CDCl$_3$) $\delta$ 7.63 (d, J = 8.0 Hz, 2H), 7.27 (d, J = 8.5 Hz, 2H), 7.22 (d, J = 7.9 Hz, 2H), 6.87 (d, J = 8.4 Hz, 2H), 5.91 (s, 1H), 2.40 (s, 3H), 1.99–1.74 (m, 9H), 1.53–1.34 (m, 1H). $^{13}$C NMR (125 MHz, CDCl$_3$) $\delta$ 179.0, 174.3, 151.3, 141.9, 129.3, 128.9, 128.3, 127.1, 126.6, 122.7, 92.3, 90.3, 34.4, 24.7, 22.2, 21.6. HRMS (ESI-MS): $m/z$ calcd for $C_{22}H_{23}ClNO$: 352.1463 [M + H]$^+$ is found to be 352.1479.

### 4.16. 4-((2-(p-tolyl)-1-oxaspiro[4.5]dec-2-en-4-ylidene) Amino)phenol (6e)

Yellow solid (248.8 mg, 91% yield). mp 222–224°C. $^1$H NMR (500 MHz, CDCl$_3$) $\delta$ 7.63 (d, J = 8.0 Hz, 2H), 7.21 (d, J = 7.9 Hz, 2H), 6.82 (d, J = 8.5 Hz, 2H), 6.75 (d, J = 8.5 Hz, 2H), 6.05 (s, 1H), 2.39 (s, 3H), 2.04–1.67 (m, 9H), 1.55–1.30 (m, 1H). $^{13}$C NMR (125 MHz, CDCl$_3$) $\delta$ 178.9, 173.8, 153.1, 143.8, 141.7, 129.3, 127.2, 126.6, 122.6, 116.1, 92.9, 90.3, 34.4, 24.6, 22.2, 21.6. HRMS (ESI-MS): $m/z$ calcd for $C_{22}H_{24}NO_2$: 334.1802 [M + H]$^+$ is found to be 334.1781.

### 4.17. N,2-diphenyl-1-oxaspiro[4.5]dec-2-en-4-imine (6f)

Yellow oil (146.8 mg, 59% yield). $^1$H NMR (500 MHz, CDCl$_3$) $\delta$ 7.74 (d, J = 6.9 Hz, 2H), 7.48–7.37 (m, 3H), 7.33 (t, J = 7.7 Hz, 2H), 7.08 (t, J = 7.4 Hz, 1H), 6.94 (d, J = 7.5 Hz, 2H), 6.01 (s, 1H), 1.96–1.78 (m, 9H), 1.50–1.43 (m, 1H). $^{13}$C NMR (125 MHz, CDCl$_3$) $\delta$ 178.2, 173.5, 152.5, 131.1, 130.0, 128.9, 128.6, 126.6, 123.2, 121.3, 93.3, 90.2, 34.5, 24.7, 22.2. HRMS (ESI-MS): $m/z$ calcd for $C_{21}H_{22}NO$: 304.1696 [M + H]$^+$ is found to be 304.1705.

### 4.18. 2-phenyl-N-(p-tolyl)-1-oxaspiro[4.5]dec-2-en-4-imine (6g)

Yellow oil (135.4 mg, 52% yield). $^1$H NMR (500 MHz, CDCl$_3$) $\delta$ 7.74 (d, J = 6.8 Hz, 2H), 7.45–7.30 (m, 3H), 7.14 (d, J = 7.9 Hz, 2H), 6.86 (d, J = 8.0 Hz, 2H), 6.05 (s, 1H), 2.35 (s, 3H), 1.90–1.78 (m, 9H), 1.50–1.37 (m, 1H). $^{13}$C NMR (125 MHz, CDCl$_3$) $\delta$ 177.9, 173.2, 149.9, 132.6, 131.0, 130.0, 129.4, 128.5, 126.5, 121.1, 93.3, 90.1, 34.4, 24.7, 22.2, 20.9. HRMS (ESI-MS): $m/z$ calcd for $C_{22}H_{24}NO$: 318.1852 [M + H]$^+$ is found to be 318.1865.

### 4.19. N-(4-methoxyphenyl)-2-phenyl-1-oxaspiro[4.5]dec-2-en-4-imine (6h)

Yellow oil (134.0 mg, 49% yield). $^1$H NMR (500 MHz, CDCl$_3$) $\delta$ 7.70 (d, $J$ = 6.7 Hz, 2H), 7.46–7.29 (m, 3H), 7.00–6.77 (m, 4H), 6.04 (s, 1H), 3.77 (s, 3H), 1.85–1.73 (m, 9H), 1.44–1.28 (m, 1H). $^{13}$C NMR (125 MHz, CDCl$_3$) $\delta$ 177.7, 173.1, 155.8, 145.7, 130.9, 130.0, 128.5, 126.4, 122.2, 114.1, 93.3, 90.1, 55.4, 34.4, 24.6, 22.1. HRMS (ESI-MS): $m/z$ calcd for C$_{22}$H$_{24}$NO$_2$: 334.1802 [M + H]$^+$ is found to be 334.1779.

### 4.20. N-(4-chlorophenyl)-2-phenyl-1-oxaspiro[4.5]dec-2-en-4-imine (6i)

Yellow oil (119.1 mg, 43% yield). $^1$H NMR (500 MHz, CDCl$_3$) $\delta$ 7.74 (d, $J$ = 7.1 Hz, 2H), 7.41–7.36 (m, 3H), 7.28 (d, $J$ = 8.5 Hz, 2H), 6.87 (d, $J$ = 8.5 Hz, 2H), 5.97 (s, 1H), 1.95–1.68 (m, 9H), 1.57–1.34 (m, 1H). $^{13}$C NMR (125 MHz, CDCl$_3$) $\delta$ 178.9, 174.0, 151.2, 131.3, 129.8, 128.9, 128.6, 128.4, 126.6, 122.6, 93.0, 90.4, 34.4, 24.6, 22.1. HRMS (ESI-MS): $m/z$ calcd for C$_{21}$H$_{21}$ClNO: 338.1306 [M + H]$^+$ is found to be 338.1292.

### 4.21. 4-((2-phenyl-1-oxaspiro[4.5]dec-2-en-4-ylidene) Amino)phenol (6j)

Yellow solid (246.2 mg, 94% yield). mp 188–190°C. $^1$H NMR (500 MHz, DMSO-d$_6$) $\delta$ 9.13 (s, 1H), 7.83 (d, $J$ = 7.3 Hz, 2H), 7.64–7.34 (m, 3H), 7.88–6.60 (m, 4H), 6.33 (s, 1H), 1.98–1.62 (m, 9H), 1.48–1.21 (m, 1H). $^{13}$C NMR (125 MHz, DMSO-d$_6$) $\delta$ 175.1, 172.0, 153.4, 143.3, 131.1, 129.3, 128.6, 126.3, 122.1, 115.3, 93.2, 89.2, 34.1, 24.2, 21.6. HRMS (ESI-MS): $m/z$ calcd for C$_{21}$H$_{22}$NO$_2$: 320.1645 [M + H]$^+$ is found to be 320.1636.

### 4.22. 2-(4-chlorophenyl)-N-phenyl-1-oxaspiro[4.5]dec-2-en-4-imine (6k)

Yellow oil (152.4 mg, 55% yield). $^1$H NMR (500 MHz, DMSO-d$_6$) $\delta$ 7.83 (d, $J$ = 8.5 Hz, 2H), 7.49 (d, $J$ = 8.5 Hz, 2H), 7.28 (t, $J$ = 7.7 Hz, 2H), 7.02 (t, $J$ = 7.4 Hz, 1H), 6.83 (d, $J$ = 7.5 Hz, 2H), 6.26 (s, 1H), 1.85–1.60 (m, 9H), 1.44–1.24 (m, 1H). $^{13}$C NMR (125 MHz, DMSO-d$_6$) $\delta$ 176.4, 171.6, 151.6, 135.9, 128.8, 128.8, 128.3, 128.0, 123.0, 120.9, 93.6, 89.5, 34.0, 24.2, 21.5. HRMS (ESI-MS): $m/z$ calcd for C$_{21}$H$_{21}$ClNO: 338.1306 [M + H]$^+$ is found to be 338.1286.

### 4.23. 2-(4-chlorophenyl)-N-(p-tolyl)-1-oxaspiro[4.5]dec-2-en-4-imine (6l)

Yellow oil (161.6 mg, 56% yield). $^1$H NMR (500 MHz, CDCl$_3$) $\delta$ 7.65 (d, $J$ = 8.5 Hz, 2H), 7.37 (d, $J$ = 8.5 Hz, 2H), 7.13 (d, $J$ = 7.9 Hz, 2H), 6.83 (d, $J$ = 8.1 Hz, 2H), 6.01 (s, 1H), 2.34 (s, 3H), 1.88–1.72 (m, 9H), 1.52–1.35 (m, 1H). $^{13}$C NMR (125 MHz, CDCl$_3$) $\delta$ 177.6, 171.9, 149.8, 137.0, 132.7, 129.5, 128.8, 128.6, 127.8, 121.1, 93.7, 90.4, 34.4, 24.6, 22.1, 20.9. HRMS (ESI-MS): $m/z$ calcd for C$_{22}$H$_{23}$ClNO: 352.1463 [M + H]$^+$ is found to be 352.1449.

### 4.24. 2-(4-chlorophenyl)-N-(4-methoxyphenyl)-1-oxaspiro[4.5]dec-2-en-4-imine (6m)

Yellow oil (178.0 mg, 59% yield). $^1$H NMR (500 MHz, CDCl$_3$) $\delta$ 7.66 (d, $J$ = 8.5 Hz, 2H), 7.37 (d, $J$ = 8.5 Hz, 2H), 7.00–6.77 (m, 4H), 6.04 (s, 1H), 3.81 (s, 3H), 1.91–1.68 (m, 9H), 1.55–1.34 (m, 1H). $^{13}$C NMR (125 MHz, CDCl$_3$) $\delta$ 177.5, 171.8, 156.0, 145.6, 137.0, 128.8, 128.6, 127.8, 122.2, 114.2, 93.6, 90.4, 55.4, 34.4, 24.6, 22.1. HRMS (ESI-MS): $m/z$ calcd for C$_{22}$H$_{23}$ClNO$_2$: 368.1412 [M + H]$^+$ is found to be 368.1400.

### 4.25. N,2-bis(4-chlorophenyl)-1-oxaspiro[4.5]dec-2-en-4-imine (6n)

Yellow solid (131.3 mg, 43% yield). mp 100–103°C. $^1$H NMR (500 MHz, CDCl$_3$) $\delta$ 7.66 (d, $J$ = 8.5 Hz, 2H), 7.39 (d, $J$ = 8.5 Hz, 2H), 7.28 (d, $J$ = 8.5 Hz, 2H), 6.86 (d, $J$ = 8.5 Hz, 2H), 5.94 (s, 1H), 1.99–1.79 (m, 9H), 1.56–1.35 (m, 1H). $^{13}$C NMR (125 MHz, CDCl$_3$) $\delta$ 178.5, 172.7, 151.0, 137.3, 129.0, 128.9, 128.5, 128.3, 127.9, 122.5, 93.3, 90.6, 34.4, 24.6, 22.1. HRMS (ESI-MS): $m/z$ calcd for C$_{21}$H$_{20}$Cl$_2$NO: 372.0917 [M + H]$^+$ is found to be 372.0900.

### 4.26. 4-((2-(4-chlorophenyl)-1-oxaspiro[4.5]dec-2-en-4-ylidene) Amino)phenol (6o)

Yellow solid (264.0 mg, 91% yield). mp 224–225°C. $^1$H NMR (500 MHz, DMSO-d$_6$) $\delta$ 7.87 (d, $J$ = 8.3 Hz, 2H), 7.53 (d, $J$ = 8.3 Hz, 2H), 6.86–6.60 (m, 4H), 6.41 (s, 1H), 1.97–1.54 (m, 9H), 1.50–1.22 (m, 1H). $^{13}$C

NMR (125 MHz, DMSO-$d_6$) $\delta$ 174.9, 170.8, 153.5, 143.2, 135.7, 128.8, 128.2, 128.1, 115.3, 93.9, 89.4, 34.1, 24.2, 21.6. HRMS (ESI-MS): *m/z* calcd for $C_{21}H_{21}ClNO_2$: 354.1255 $[M + H]^+$ is found to be 354.1245.

## 4.27. *N*,2,2-triphenyl-5-(*p*-tolyl)furan-3(2*H*)-imine (6p)

Yellow solid (135.0 mg, 41% yield). mp 151–153°C. $^1$H NMR (500 MHz, CDCl$_3$) $\delta$ 7.89–7.68 (m, 6H), 7.53–7.41 (m, 8H), 7.30 (d, *J* = 7.9 Hz, 2H), 7.17 (t, *J* = 7.3 Hz, 1H), 7.06 (d, *J* = 7.6 Hz, 2H), 6.21 (s, 1H), 2.45 (s, 3H). $^{13}$C NMR (125 MHz, CDCl$_3$) $\delta$ 175.2, 173.6, 152.3, 142.0, 141.2, 129.3, 128.9, 128.1, 128.0, 127.1, 126.6, 126.5, 123.4, 120.9, 93.6, 92.4, 21.5. HRMS (ESI-MS): *m/z* calcd for $C_{29}H_{24}NO$: 402.1852 $[M + H]^+$ is found to be 402.1869.

## 4.28. 2,2-diphenyl-*N*,5-di-*p*-tolylfuran-3(2*H*)-imine (6q)

Yellow solid (177.1 mg, 52% yield). mp 141–143°C. $^1$H NMR (500 MHz, CDCl$_3$) $\delta$ 7.75–7.56 (m, 6H), 7.42–7.23 (m, 6H), 7.18 (d, *J* = 8.1 Hz, 2H), 7.08 (d, *J* = 7.9 Hz, 2H), 6.82 (d, *J* = 8.0 Hz, 2H), 6.09 (s, 1H), 2.34 (s, 3H), 2.29 (s, 3H). $^{13}$C NMR (125 MHz, CDCl$_3$) $\delta$ 175.0, 173.4, 149.7, 141.9, 141.3, 132.8, 129.4, 129.4, 128.1, 128.0, 127.2, 126.7, 126.6, 120.9, 93.7, 92.4, 21.6, 20.9. HRMS (ESI-MS): *m/z* calcd for $C_{30}H_{26}NO$: 416.2009 $[M + H]^+$ is found to be 416.2028.

## 4.29. *N*-(4-methoxyphenyl)-2,2-diphenyl-5-(*p*-tolyl)furan-3(2*H*)-imine (6r)

Yellow solid (173.4 mg, 49% yield). mp 140–142°C. $^1$H NMR (500 MHz, CDCl$_3$) $\delta$ 7.79–7.56 (m, 6H), 7.41–7.11 (m, 8H), 6.94–6.71 (m, 4H), 6.13 (s, 1H), 3.75 (s, 3H), 2.34 (s, 3H). $^{13}$C NMR (125 MHz, CDCl$_3$) $\delta$ 174.9, 173.3, 156.1, 145.6, 141.9, 141.3, 129.4, 128.1, 127.9, 127.2, 126.6, 126.6, 122.1, 114.1, 93.6, 92.4, 55.4, 21.6. HRMS (ESI-MS): *m/z* calcd for $C_{30}H_{26}NO_2$: 432.1958 $[M + H]^+$ is found to be 432.1974.

## 4.30. *N*-(4-chlorophenyl)-2,2-diphenyl-5-(*p*-tolyl)furan-3(2*H*)-imine (6s)

Yellow solid (121.5 mg, 34% yield). mp 148–150°C. $^1$H NMR (500 MHz, CDCl$_3$) $\delta$ 7.87–7.56 (m, 6H), 7.45–7.08 (m, 10H), 6.90 (d, *J* = 8.5 Hz, 2H), 6.08 (s, 1H), 2.40 (s, 3H). $^{13}$C NMR (125 MHz, CDCl$_3$) $\delta$ 176.0, 174.3, 150.9, 142.3, 140.9, 129.4, 128.9, 128.6, 128.2, 128.1, 127.1, 126.7, 126.4, 122.3, 93.3, 92.6, 21.6. HRMS (ESI-MS): *m/z* calcd for $C_{29}H_{23}ClNO$: 436.1463 $[M + H]^+$ is found to be 436.1464.

## 4.31. 4-((2,2-diphenyl-5-(*p*-tolyl)furan-3(2*H*)-ylidene) amino)phenol (6t)

Yellow solid (191.7 mg, 56% yield). mp 210–212°C. $^1$H NMR (500 MHz, DMSO-$d_6$) $\delta$ 9.22 (s, 1H), 7.87 (d, *J* = 8.1 Hz, 2H), 7.59 (d, *J* = 7.4 Hz, 4H), 7.41–7.20 (m, 8H), 6.91–6.75 (m, 4H), 6.50 (s, 1H), 2.36 (s, 3H). $^{13}$C NMR (125 MHz, DMSO-$d_6$) $\delta$ 172.5, 172.4, 153.8, 142.8, 141.8, 141.2, 129.4, 128.1, 127.9, 126.5, 126.5, 125.9, 122.1, 115.4, 93.6, 91.4, 21.0. HRMS (ESI-MS): *m/z* calcd for $C_{29}H_{24}NO_2$: 418.1802 $[M + H]^+$ is found to be 418.1817.

An efficient method to synthesize furan-3(2*H*)-imine derivatives was established by reaction between the $\alpha,\beta$-unsaturated ketones and aniline derivatives.

Data accessibility. This article has no additional data.

**Authors' contributions.** A.J.R. and I.A.S. contributed equally to this work. A.J.R contributed in designing, performing and supervising the experimental work. I.A.S contributed in analysis of data and compiling the manuscript. E.A.Y. and D.M.M.J. analysed the NMR and contributed in revising the manuscript. R.A.A. contributed in designing and supervising the experiments and writing the manuscript.

**Competing interests.** We declare we have no competing interests.

**Funding.** The authors would acknowledge the generous funding from Deanship of Scientific Research, Al-Balqa Applied University, grant no. 864/2019/2020 and the Center of Advanced Material/RISE at the University of Sharjah, UAE, for logistic support.

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
