## [Peer Review File · Royal Society Open Science]

Review History

RSOS-211145.R0 (Original submission)

Review form: Reviewer 1

Is the manuscript scientifically sound in its present form?

Yes

Are the interpretations and conclusions justified by the results?

Yes

Is the language acceptable?

No

Do you have any ethical concerns with this paper?

No

Have you any concerns about statistical analyses in this paper?

No

Recommendation?

Accept with minor revision (please list in comments)

Comments to the Author(s)

Review of “An efficient synthesis of furan-3(2H)-imine scaffold from alkyones.”

AlQawasmeh and coworkers describe the intramolecular condensation of and anilines and hydroxybutynones to yield butenolide-imines. Based on a literature search, this appears to be a novel coupling procedure. Included in the manuscript is (a) an introduction/introductory figure which highlights some related methods to synthesize butenolide-imines, (b) a description of how the hydroxybutynones are prepared (Scheme 2), and the scope of the condensation reaction (Table 1). Scheme 3 shows that you cannot make these molecules by condensing aniline and butenolide. Scheme 4 provides a reasonable electron-push mechanism for the process.

- Is there a generally accepted common name for “butenolide-imines” functional group (this term does not appear in the paper, nor do other terms; in references below, I also see the “lactone-imine” “iminolactones” and furanimine). Consider adding a name or listing ways this functional group has been named previously.
- The manuscript figures and flow is outlined reasonably. However, this reviewer does not prefer how the data in Table 1 is shown. If there is space, I strongly recommend actually showing the structures. It will be easier to understand, view, and appreciate the scope.
- While the general outline and flow is acceptable, the manuscript is in need of significant copy editing prior to publication: there are grammatical errors throughout the manuscript.
- While searching for other ways to make “butenolide-imines” I came across some methods that are not cited. Considering this is a methodology paper, the authors should provide a more-representative analysis of other ways to make this motif.
 - o (1) Nedolya, N. A.; Schlyakhtina, N. I.; Zinov'eva, V. P.; Albanov, A. I.; Brandsma, L. Synthesis and Ag⁺-Catalyzed Cyclization of 2,3-Dienamides. *Tetrahedron Lett.* 2002, 43 (8), 1569–1571. [https://doi.org/10.1016/S0040-4039\(02\)00017-5](https://doi.org/10.1016/S0040-4039(02)00017-5).
 - o (2) Ma, S.; Xie, H. Steric Hindrance-Controlled Pd(0)-Catalyzed Coupling-Cyclization of 2,3-Allenamides and Organic Iodides. An Efficient Synthesis of Iminolactones and γ -Hydroxy- γ -Lactams. *J. Org. Chem.* 2002, 67 (18), 6575–6578. <https://doi.org/10.1021/jo025967v>.
 - o (3) Ma, S.; Xie, H. Study on Halolactamization- γ -Hydroxylation or Haloiminolactonization of 2,3-Alkadienamides. *Tetrahedron* 2005, 61 (1), 251–258. <https://doi.org/10.1016/j.tet.2004.10.050>.
 - o (4) Wang, C.; Lu, J.; Mao, G.; Xi, Z. Preparation and Electrophilic Cyclization of Multisubstituted Dienamides Leading to Cyclic Iminoethers. *J. Org. Chem.* 2005, 70 (13), 5150–5156. <https://doi.org/10.1021/jo050433q>.
 - o (5) Ma, S.; Gu, Z.; Yu, Z. Pd(CH₃CN)₂Cl₂-Catalyzed Oxidative Heterodimerization Reaction of 2,3-Allenamides and 1,2-Allenyl Ketones: An Efficient Synthesis of 4-(Furan-3'-Y1)-2(5H)-Furanimines. *J. Org. Chem.* 2005, 70 (16), 6291–6294. <https://doi.org/10.1021/jo0507441>.
 - o (6) Tang, Y.; Li, C. Facile 5-Endo Electrophilic Cyclization of Unsaturated Amides with TBuOCl/12. *Tetrahedron Lett.* 2006, 47 (23), 3823–3825. <https://doi.org/10.1016/j.tetlet.2006.03.166>.
 - o (7) Chen, G.; Li, Y.; Fu, C.; Ma, S. Homodimeric Coupling-Cyclization Reaction of 2,3-Allenamides. *Tetrahedron* 2009, 65 (23), 4547–4550. <https://doi.org/10.1016/j.tet.2009.03.083>.
 - o (8) Chen, G.; Zhang, Y.; Fu, C.; Ma, S. A Facile Synthesis of β -Allenyl Furanimines via Pd-Catalyzed Cyclization of 2,3-Allenamides with Propargylic Carbonates. *Tetrahedron* 2011, 67 (12), 2332–2337. <https://doi.org/10.1016/j.tet.2011.01.041>.
 - o (9) Li, J.; Liu, Y.; Li, C.; Jie, H.; Jia, X. Atom-Economical Synthesis of the Functionalized Spirocyclic Oxindole-Butenolide via Three-Component [2 + 2 + 1] Cycloaddition Strategy. *Green Chem.* 2012, 14 (5), 1314–1321. <https://doi.org/10.1039/c2gc00015f>.

- o (10) Terzidis, M. A.; Zarganes-Tzitzikas, T.; Tsimenidis, C.; Stephanidou-Stephanatou, J.; Tsoleridis, C. A.; Kostakis, G. E. One-Pot Five-Component Synthesis of Spirocyclopenta[b]Chromene Derivatives and Their Acid-Catalyzed Rearrangement. *J. Org. Chem.* 2012, 77 (20), 9018–9028. <https://doi.org/10.1021/jo3014947>.
- o (11) Asghari, S.; Tayebi, Z.; Khabbazi Habibi, A. Reaction of Tert-Butyl Isocyanide with Alkyl Propiolates in the Presence of α -Chloro Ketones. *J. Heterocycl. Chem.* 2013, 50 (4), 874–878. <https://doi.org/10.1002/jhet.1713>.
- The references are formatted incorrectly, I believe.

Review form: Reviewer 2

Is the manuscript scientifically sound in its present form?

Yes

Are the interpretations and conclusions justified by the results?

Yes

Is the language acceptable?

Yes

Do you have any ethical concerns with this paper?

No

Have you any concerns about statistical analyses in this paper?

No

Recommendation?

Accept with minor revision (please list in comments)

Comments to the Author(s)

This manuscript describes the synthesis of a range of novel furan3(2H)-imine derivatives and, after the corrections listed below, I believe is suitable for publication.

The synthesis of the substrates is routine chemistry and would be better as an appendix or in the SI according to the conventions of this journal.

Table 1 is very hard to read because there are 4 R groups. I suggest that it is presented in a format where the general reaction is depicted at the top and then every product is drawn out with and number and the yield and reaction time underneath. This way the reader actually gets to see all the different products.

"However, there was no significant amount of the target product observed in these reactions." - this is in relation to Scheme 3 which shows that you can't make the product from the corresponding vinylogous ester. Did any other products form here though or was there no reaction at all? Should state which.

The proposed mechanism features an intramolecular proton transfer - this is more likely to be intermolecular.

The addition of aniline to 4a is described in the manuscript as a Michael addition but, strictly speaking, Michael addition refers to the conjugate addition of an enolate nucleophile so 1,4-addition or conjugate addition is a better term to use.

The geometry of the C=N double bond in the products has been assumed based on steric effects which I think is reasonable but the authors should probably state this in the text somewhere.

In the experimental section

ml should be mL

SiO₂ the 2 should always be subscript

J values for mutually coupled Hs should match each other

R_f values should be given for those compounds that have been purified by column chromatography

Decision letter (RSOS-211145.R0)

Dear Dr Shehadi:

Title: An efficient synthesis of furan-3(2H)-imine scaffold from alkynones

Manuscript ID: RSOS-211145

Thank you for submitting the above manuscript to Royal Society Open Science. On behalf of the Editors and the Royal Society of Chemistry, I am pleased to inform you that your manuscript will be accepted for publication in Royal Society Open Science subject to minor revision in accordance with the referee suggestions. Please find the reviewers' comments at the end of this email.

The reviewers and handling editors have recommended publication, but also suggest some minor revisions to your manuscript. Therefore, I invite you to respond to the comments and revise your manuscript.

Because the schedule for publication is very tight, it is a condition of publication that you submit the revised version of your manuscript before 29-Sep-2021. Please note that the revision deadline will expire at 00.00am on this date. If you do not think you will be able to meet this date please let me know immediately.

Kind regards,
Dr Ellis Wilde
Publishing Editor, Journals

On behalf of the Subject Editor Professor Anthony Stace and the Associate Editor Dr Andrew Harned.

RSC Associate Editor
Comments to the Author:

The authors report an interesting method to prepare an under-explored class of compounds. The reported method is straightforward and has a reasonable demonstrated scope. The referees have, however, raised several relatively minor suggestions that would strengthen the paper as a whole.

I would like to point out that Reviewer 2 suggest moving Scheme 2 to the supporting information. As this is a general journal, I am fine if the authors choose to leave it as part of the

main article. This does serve a purpose in showing the facile nature in which the final compounds can be prepared.

RSC Subject Editor

Comments to the Author:

(There are no comments.)

Reviewer comments to Author:

Reviewer: 1

Comments to the Author(s)

Review of "An efficient synthesis of furan-3(2H)-imine scaffold from alkyones."

AlQawasmeh and coworkers describe the intramolecular condensation of anilines and hydroxybutynones to yield butenolide-imines. Based on a literature search, this appears to be a novel coupling procedure. Included in the manuscript is (a) an introduction/introductory figure which highlights some related methods to synthesize butenolide-imines, (b) a description of how the hydroxybutynones are prepared (Scheme 2), and the scope of the condensation reaction (Table 1). Scheme 3 shows that you cannot make these molecules by condensing aniline and butenolide. Scheme 4 provides a reasonable electron-push mechanism for the process.

- Is there a generally accepted common name for "butenolide-imines" functional group (this term does not appear in the paper, nor do other terms; in references below, I also see the "lactone-imine" "iminolactones" and furanimine). Consider adding a name or listing ways this functional group has been named previously.

- The manuscript figures and flow is outlined reasonably. However, this reviewer does not prefer how the data in Table 1 is shown. If there is space, I strongly recommend actually showing the structures. It will be easier to understand, view, and appreciate the scope.

- While the general outline and flow is acceptable, the manuscript is in need of significant copy editing prior to publication: there are grammatical errors throughout the manuscript.

- While searching for other ways to make "butenolide-imines" I came across some methods that are not cited. Considering this is a methodology paper, the authors should provide a more-representative analysis of other ways to make this motif.

o (1) Nedolya, N. A.; Schlyakhtina, N. I.; Zinov'eva, V. P.; Albanov, A. I.; Brandsma, L. Synthesis and Ag⁺-Catalyzed Cyclization of 2,3-Dienamides. *Tetrahedron Lett.* 2002, 43 (8), 1569-1571. [https://doi.org/10.1016/S0040-4039\(02\)00017-5](https://doi.org/10.1016/S0040-4039(02)00017-5).

o (2) Ma, S.; Xie, H. Steric Hindrance-Controlled Pd(0)-Catalyzed Coupling-Cyclization of 2,3-Allenamides and Organic Iodides. An Efficient Synthesis of Iminolactones and γ -Hydroxy- γ -Lactams. *J. Org. Chem.* 2002, 67 (18), 6575-6578. <https://doi.org/10.1021/jo025967v>.

o (3) Ma, S.; Xie, H. Study on Halolactamization- γ -Hydroxylation or Haloiminolactonization of 2,3-Alkadienamides. *Tetrahedron* 2005, 61 (1), 251-258. <https://doi.org/10.1016/j.tet.2004.10.050>.

o (4) Wang, C.; Lu, J.; Mao, G.; Xi, Z. Preparation and Electrophilic Cyclization of Multisubstituted Dienamides Leading to Cyclic Iminoethers. *J. Org. Chem.* 2005, 70 (13), 5150-5156. <https://doi.org/10.1021/jo050433q>.

o (5) Ma, S.; Gu, Z.; Yu, Z. Pd(CH₃CN)₂Cl₂-Catalyzed Oxidative Heterodimerization Reaction of 2,3-Allenamides and 1,2-Allenyl Ketones: An Efficient Synthesis of 4-(Furan-3'-Yl)-2(5H)-Furanimines. *J. Org. Chem.* 2005, 70 (16), 6291-6294. <https://doi.org/10.1021/jo0507441>.

o (6) Tang, Y.; Li, C. Facile 5-Endo Electrophilic Cyclization of Unsaturated Amides with TBuOCl/I₂. *Tetrahedron Lett.* 2006, 47 (23), 3823-3825. <https://doi.org/10.1016/j.tetlet.2006.03.166>.

o (7) Chen, G.; Li, Y.; Fu, C.; Ma, S. Homodimeric Coupling-Cyclization Reaction of 2,3-Allenamides. *Tetrahedron* 2009, 65 (23), 4547-4550. <https://doi.org/10.1016/j.tet.2009.03.083>.

- o (8) Chen, G.; Zhang, Y.; Fu, C.; Ma, S. A Facile Synthesis of β -Allenyl Furanimines via Pd-Catalyzed Cyclization of 2,3-Allenamides with Propargylic Carbonates. *Tetrahedron* 2011, 67 (12), 2332–2337. <https://doi.org/10.1016/j.tet.2011.01.041>.
- o (9) Li, J.; Liu, Y.; Li, C.; Jie, H.; Jia, X. Atom-Economical Synthesis of the Functionalized Spirocyclic Oxindole-Butenolide via Three-Component [2 + 2 + 1] Cycloaddition Strategy. *Green Chem.* 2012, 14 (5), 1314–1321. <https://doi.org/10.1039/c2gc00015f>.
- o (10) Terzidis, M. A.; Zarganes-Tzitzikas, T.; Tsimenidis, C.; Stephanidou-Stephanatou, J.; Tsoleridis, C. A.; Kostakis, G. E. One-Pot Five-Component Synthesis of Spirocyclopenta[b]Chromene Derivatives and Their Acid-Catalyzed Rearrangement. *J. Org. Chem.* 2012, 77 (20), 9018–9028. <https://doi.org/10.1021/jo3014947>.
- o (11) Asghari, S.; Tayebi, Z.; Khabbazi Habibi, A. Reaction of Tert-Butyl Isocyanide with Alkyl Propiolates in the Presence of α -Chloro Ketones. *J. Heterocycl. Chem.* 2013, 50 (4), 874–878. <https://doi.org/10.1002/jhet.1713>.

- The references are formatted incorrectly, I believe.

Reviewer: 2

Comments to the Author(s)

This manuscript describes the synthesis of a range of novel furan3(2H)-imine derivatives and, after the corrections listed below, I believe is suitable for publication.

The synthesis of the substrates is routine chemistry and would be better as an appendix or in the SI according to the conventions of this journal.

Table 1 is very hard to read because there are 4 R groups. I suggest that it is presented in a format where the general reaction is depicted at the top and then every product is drawn out with and number and the yield and reaction time underneath. This way the reader actually gets to see all the different products.

"However, there was no significant amount of the target product observed in these reactions." - this is in relation to Scheme 3 which shows that you can't make the product from the corresponding vinylogous ester. Did any other products form here though or was there no reaction at all? Should state which.

The proposed mechanism features an intramolecular proton transfer - this is more likely to be intermolecular.

The addition of aniline to 4a is described in the manuscript as a Michael addition but, strictly speaking, Michael addition refers to the conjugate addition of an enolate nucleophile so 1,4-addition or conjugate addition is a better term to use.

The geometry of the C=N double bond in the products has been assumed based on steric effects which I think is reasonable but the authors should probably state this in the text somewhere.

In the experimental section

ml should be mL

SiO₂ the 2 should always be subscript

J values for mutually coupled Hs should match each other

R_f values should be given for those compounds that have been purified by column chromatography

Author's Response to Decision Letter for (RSOS-211145.R0)

See Appendix A.

Decision letter (RSOS-211145.R1)

Dear Dr Shehadi:

Title: An efficient synthesis of furan-3(2H)-imine scaffold from alkynones
Manuscript ID: RSOS-211145.R1

It is a pleasure to accept your manuscript in its current form for publication in Royal Society Open Science. The chemistry content of Royal Society Open Science is published in collaboration with the Royal Society of Chemistry.

Yours sincerely,
Dr Ellis Wilde
Publishing Editor, Journals

On behalf of the Subject Editor Professor Anthony Stace and the Associate Editor Dr Andrew Harned.

RSC Associate Editor
Comments to the Author:

The authors have provided suitable responses to all of the referee comments. I can recommend publication at this time.

Reviewer(s)' Comments to Author:

Appendix A

To: Associate editor of the Royal Society Open Science

From: Dr. Ihsan Shehadi, Corresponding author of Manuscript ID RSOS-211145, on behalf of all authors.

Subject: Responses to reviewers' queries

Date: September 23, 2021

Dear Associate editor,

Thank you for your efforts in processing our manuscript to be considered for publication in the esteemed journal "Royal Society Open Science". In addition, we would like to express our great delight in the reviewers' feedback as they added more value to the final version of the manuscript.

Please find below our detailed responses to the reviewers' queries:

Responses to Reviewer: 1

Comments to the Author(s)

Review of "An efficient synthesis of furan-3(2H)-imine scaffold from alkynones."

AlQawasmeh and coworkers describe the intramolecular condensation of anilines and hydroxybutynones to yield butenolide-imines. Based on a literature search, this appears to be a novel coupling procedure. Included in the manuscript is (a) an introduction/introductory figure which highlights some related methods to synthesize butenolide-imines, (b) a description of how the hydroxybutynones are prepared (Scheme 2), and the scope of the condensation reaction (Table 1). Scheme 3 shows that you cannot make these molecules by condensing aniline and butenolide. Scheme 4 provides a reasonable electron-push mechanism for the process.

- Is there a generally accepted common name for "butenolide-imines" functional group (this term does not appear in the paper, nor do other terms; in references below, I also see the "lactone-imine" "iminolactones" and furanimine). Consider adding a name or listing ways this functional group has been named previously.

Response: We use the term "furan-3(2H)-imine" many times in Title, abstract, introduction and result and discussion. This is the correct terminology adopted for this scaffold and it is used by Chen *et al* (reference 14).

- The manuscript figures and flow is outlined reasonably. However, this reviewer does not prefer how the data in Table 1 is shown. If there is space, I strongly recommend actually showing the structures. It will be easier to understand, view, and appreciate the scope.

Response: All structures are drawn and presented in table 1.

- While the general outline and flow is acceptable, the manuscript is in need of significant copy editing prior to publication: there are grammatical errors throughout the manuscript.

Response: A concise editorial review of the text has been conducted and all grammatical disputes are addressed.

- While searching for other ways to make "butenolide-imines" I came across some methods that are not cited. Considering this is a methodology paper, the authors should provide a more-representative analysis of other ways to make this motif.

o (1) Nedolya, N. A.; Schlyakhtina, N. I.; Zinov'eva, V. P.; Albanov, A. I.; Brandsma, L. Synthesis and Ag⁺-Catalyzed Cyclization of 2,3-Dienamides. *Tetrahedron Lett.* 2002, 43 (8), 1569–1571.

[https://doi.org/10.1016/S0040-4039\(02\)00017-5](https://doi.org/10.1016/S0040-4039(02)00017-5).

o (2) Ma, S.; Xie, H. Steric Hindrance-Controlled Pd(0)-Catalyzed Coupling-Cyclization of 2,3-Allenamides and Organic Iodides. An Efficient Synthesis of Iminolactones and γ -Hydroxy- γ -Lactams. *J. Org. Chem.* 2002, 67 (18), [6575–6578](https://doi.org/10.1021/jo025967v). <https://doi.org/10.1021/jo025967v>.

o (3) Ma, S.; Xie, H. Study on Halolactamization- γ -Hydroxylation or Haloiminolactonization of 2,3-Alkadienamides. *Tetrahedron* 2005, 61 (1), 251–258. <https://doi.org/10.1016/j.tet.2004.10.050>.

o (4) Wang, C.; Lu, J.; Mao, G.; Xi, Z. Preparation and Electrophilic Cyclization of Multisubstituted Dienamides Leading to Cyclic Iminoethers. *J. Org. Chem.* 2005, 70 (13), [5150–5156](https://doi.org/10.1021/jo050433q). <https://doi.org/10.1021/jo050433q>.

o (5) Ma, S.; Gu, Z.; Yu, Z. Pd(CH₃CN)₂Cl₂-Catalyzed Oxidative Heterodimerization Reaction of 2,3-Allenamides and 1,2-Allenyl Ketones: An Efficient Synthesis of 4-(Furan-3'-yl)-2(5H)-Furanimines. *J. Org. Chem.* 2005, 70 (16), [6291–6294](https://doi.org/10.1021/jo0507441). <https://doi.org/10.1021/jo0507441>.

o (6) Tang, Y.; Li, C. Facile 5-Endo Electrophilic Cyclization of Unsaturated Amides with TBuOCl/I₂. *Tetrahedron Lett.* 2006, 47 (23), [3823–3825](https://doi.org/10.1016/j.tetlet.2006.03.166). <https://doi.org/10.1016/j.tetlet.2006.03.166>.

o (7) Chen, G.; Li, Y.; Fu, C.; Ma, S. Homodimeric Coupling-Cyclization Reaction of 2,3-Allenamides. *Tetrahedron* 2009, 65 (23), [4547–4550](https://doi.org/10.1016/j.tet.2009.03.083). <https://doi.org/10.1016/j.tet.2009.03.083>.

o (8) Chen, G.; Zhang, Y.; Fu, C.; Ma, S. A Facile Synthesis of β -Allenyl Furanimines via Pd-Catalyzed Cyclization of 2,3-Allenamides with Propargylic Carbonates. *Tetrahedron* 2011, 67 (12), [2332–2337](https://doi.org/10.1016/j.tet.2011.01.041). <https://doi.org/10.1016/j.tet.2011.01.041>.

o (9) Li, J.; Liu, Y.; Li, C.; Jie, H.; Jia, X. Atom-Economical Synthesis of the Functionalized Spirocyclic Oxindole-Butenolide via Three-Component [2 + 2 + 1] Cycloaddition Strategy. *Green Chem.* 2012, 14 (5), 1314–1321. <https://doi.org/10.1039/c2gc00015f>.

o (10) Terzidis, M. A.; Zarganes-Tzitzikas, T.; Tsimenidis, C.; Stephanidou-Stephanatou, J.; Tsoleridis, C. A.; Kostakis, G. E. One-Pot Five-Component Synthesis of Spirocyclopenta[b]Chromene Derivatives and Their Acid-Catalyzed Rearrangement. *J. Org. Chem.* 2012, 77 (20), [9018–9028](https://doi.org/10.1021/jo3014947). <https://doi.org/10.1021/jo3014947>.

o (11) Asghari, S.; Tayebi, Z.; Khabbazi Habibi, A. Reaction of Tert-Butyl Isocyanide with Alkyl Propiolates in the Presence of α -Chloro Ketones. *J. Heterocycl. Chem.* 2013, 50 (4), 874–878. <https://doi.org/10.1002/jhet.1713>.

Response: 10 references from the above are added to the paper, even though some of the mentioned references do not fall within the scope of the presented research as they address

mostly the lactone imine. However, we believe such references will add a wider scope to our research work.

- The references are formatted incorrectly, I believe.

Response: The references are fit to the journal style.

Responses to Reviewer: 2

Comments to the Author(s)

This manuscript describes the synthesis of a range of novel furan3(2H)-imine derivatives and, after the corrections listed below, I believe is suitable for publication.

- The synthesis of the substrates is routine chemistry and would be better as an appendix or in the SI according to the conventions of this journal.

Response: I prefer to leave it as part of the main article. This will serve a purpose of the pioneer work in showing the facile nature and diversity of the final products.

- Table 1 is very hard to read because there are 4 R groups. I suggest that it is presented in a format where the general reaction is depicted at the top and then every product is drawn out with and number and the yield and reaction time underneath. This way the reader actually gets to see all the different products.

Response: All structures are drawn and presented in table 1 where the R groups are actually presented.

- "However, there was no significant amount of the target product observed in these reactions." - this is in relation to Scheme 3 which shows that you can't make the product from the corresponding vinylogous ester. Did any other products form here though or was there no reaction at all? Should state which.

Response: A statement is added to indicate the absence of any product as "However, there is not any significant amount of the target product being observed in such reaction, hence, the suggested reaction pathway is not favored"

- The proposed mechanism features an intramolecular proton transfer - this is more likely to be intermolecular.

Response: we just mention it is "hydrogen transfer", without the word inter or intra as we do not have the sufficient evidence that the hydrogen transfer occurs *via* the intra or inter molecular transfer. Hence, for the sake of scientific integrity, we refrain from specifying either one.

- The addition of aniline to 4a is described in the manuscript as a Michael addition but, strictly speaking, Michael addition refers to the conjugate addition of an enolate nucleophile so 1,4-addition or conjugate addition is a better term to use.

Response: Michael addition replaced with 1,4-addition.

- The geometry of the C=N double bond in the products has been assumed based on steric effects which I think is reasonable but the authors should probably state this in the text somewhere.

Response: this sentence was added to the text "The geometry of the C=N double bonds in the products have been concluded based on the associated steric effects."

- In the experimental section,

ml should be mL

SiO₂ the 2 should always be subscript

Response: done.

- J values for mutually coupled Hs should match each other

Response: This is true, but sometimes for closed or overlapped peaks small error can be observed.

- R_f values should be given for those compounds that have been purified by column chromatography

Response: I think the R_f value can be given for any compound regardless of whether it is purified by column chromatography or not. In this line of work, we are interested in monitoring the chemical reactions and therefore purifying without any significance of reporting the R_f values.

Thank you for your care and attention,

Ihsan Shehadi,

Corresponding author of Manuscript ID RSOS-211145